# The response of Dual-leucine zipper kinase (DLK) to nocodazole: Evidence for a homeostatic cytoskeletal repair mechanism

Laura DeVault[1], Chase Mateusiak[2,3], John Palucki[1], Michael Brent[2,3], Jeffrey Milbrandt[2,4], Aaron DiAntonio[1,4]*

1 Department of Developmental Biology, Washington University School of Medicine, St. Louis, Missouri, United States of America, 2 Department of Genetics, Washington University School of Medicine, St. Louis, Missouri, United States of America, 3 Department of Computer Science & Engineering, Washington University, St. Louis, MO, United States of America, 4 Needleman Center for Neurometabolism and Axonal Therapeutics, Washington University School of Medicine, St. Louis, Missouri, United States of America

* diantonio@wustl.edu

## Abstract

Genetic and pharmacological perturbation of the cytoskeleton enhances the regenerative potential of neurons. This response requires Dual-leucine Zipper Kinase (DLK), a neuronal stress sensor that is a central regulator of axon regeneration and degeneration. The damage and repair aspects of this response are reminiscent of other cellular homeostatic systems, suggesting that a cytoskeletal homeostatic response exists. In this study, we propose a framework for understanding DLK mediated neuronal cytoskeletal homeostasis. We demonstrate that low dose nocodazole treatment activates DLK signaling. Activation of DLK signaling results in a DLK-dependent transcriptional signature, which we identify through RNA-seq. This signature includes genes likely to attenuate DLK signaling while simultaneously inducing actin regulating genes. We identify alterations to the cytoskeleton including actin-based morphological changes to the axon. These results are consistent with the model that cytoskeletal disruption in the neuron induces a DLK-dependent homeostatic mechanism, which we term the Cytoskeletal Stress Response (CSR) pathway.

## Introduction

Injury to peripheral axons leads to cytoskeletal disassembly, axon transport disruption, and axon retraction. However, injury also stimulates repair processes that result in axon regeneration [1,2]. While cytoskeletal disassembly is a feature of the early stages of peripheral neuron injury, cytoskeletal reassembly is a hallmark of the later stages [3,4]. One central participant in the axon regeneration program is Dual-Leucine Zipper Kinase (DLK), which is activated by injury and other neuronal stressors [5–8]. It has been hypothesized that DLK regulates an axonal stress response [9]. While the axon is an important locus of DLK action, this response is may be neuron-wide since whole neuron treatment with cytoskeletal disrupting drugs exerts

**Funding:** LD: F32NS117784 (national institute of neurological disorders and stroke) AD, JM:

R01NS087632 (national institute of neurological disorders and stroke) AD: R37NS065053 (national institute of neurological disorders and stroke) MB: GM 141012 (national institute general of medical sciences). The funders had no role in study design, data collection and analysis, decision to publish, or preparation of the manuscript.

**Competing interests:** I have read the journal's policy and the authors of this manuscript have the following competing interests. This does not alter our adherence to PLOS ONE policies on sharing data and materials.

greater DLK-dependent effects than axonal compartmentalized treatments [10] and DLK signaling dramatically impacts dendritic morphology [11].

DLK-dependent signaling cascades occur after physical injury of the axon, deprivation of nerve growth factor (NGF), and treatment with low doses of either the microtubule destabilizer nocodazole or the actin destabilizer cytochalasin D [10,12]. Each of these treatments disrupts the cytoskeleton. Nocodazole and cytochalasin D directly impact the microtubule or actin cytoskeleton, while physical injury and NGF deprivation impact the cytoskeleton through microtubule fragmentation and actin loss [13,14].

While DLK activation is transiently induced during development or injury, constitutive DLK signaling is observed upon genetic perturbation of cytoskeletal components and regulators. In *C. elegans*, loss of β-spectrin, a core component of the neuronal membrane periodic structure, increases neuronal susceptibility to mechanical strain and activates DLK signaling [5]. Similarly, in *Drosophila*, DLK signaling is constitutively activated by loss of Spectraplakin, a linker of the actin and microtubule cytoskeleton, Patronin, a minus-end microtubule capping protein, and Rae1, a microtubule binding protein [15–18]. Loss of Spectraplakin and Patronin each increase cytoskeletal instability in the axon [16,18,19]. These findings collectively suggest that cytoskeletal instability activates DLK signaling.

Upon activation, DLK initiates a MAP kinase signaling cascade. DLK's major direct phosphorylation target is MKK4, which in turn phosphorylates JNK [5,20]. In the axon, DLK activation triggers the retrograde transport of the DLK/MKK4/JNK complex to the cell body, leading to phosphorylation of the transcription factor cJun [7,8,12,21]. Arrival of the injury signal at the nucleus results in a suite of transcriptional changes, including the induction of cJun and other regeneration-associated genes [22–24]. These transcriptional changes promote axon outgrowth, a process driven by coordinated interaction of microtubules, actin and microfilaments. These genes, including cJun, are transiently expressed. Within days, peripheral injured neurons *in vitro* return to a basal state[25].

A homeostatic response is a regulatory process in which a disruption triggers a mechanism that returns the system to the basal state. In this work, we evaluate the hypothesis that a DLK-dependent cytoskeletal homeostatic network exists. We propose the following criteria: (1) cytoskeletal perturbation must initiate a DLK signaling cascade resulting in (2) transcriptional changes which include (3) genes capable of attenuating the strength of the DLK signaling cascade and (4) genes capable of modulating the cytoskeleton, thus resulting in (5) changes to the cytoskeleton. To investigate the hypothesis that such a cytoskeletal homeostatic network exists, we test the impact of low dose nocodazole treatment on cultured dorsal root ganglia (DRG) sensory neurons. Our findings demonstrate that cytoskeletal perturbation activates DLK signaling and generates a DLK-dependent transcriptional signature. This transcriptional signature includes both cytoskeletal regulators and MAP kinase signal attenuators. Moreover, inhibiting this DLK pathway exacerbates microtubule defects induced by low dose nocodazole. These results are consistent with the model that, upon cytoskeletal disruption, neurons engage a DLK-dependent homeostatic mechanism that we term the Cytoskeletal Stress Response pathway (CSR).

## Materials and methods

### Primary embryonic DRG culture

Primary DRGs were cultured from E13.5 CD1 mouse embryos (Charles River Laboratories). DRGs were dissected in DMEM. Once isolated, DRGs were incubated in 0.5% trypsin, 0.05% EDTA for 13 minutes. Neurons were resuspended in growth media (Neurobasal (Gibco), 2% B27 (Invtrogen), 50 ng/ml nerve growth factor (Harlan Laboratories), 1 μM 5-fluoro-2'-

deoxyuridine (Millipore Sigma), 1 μM uridine (Millipore Sigma), and penicillin/ streptomycin (Thermo Fisher Scientific)) and spotted onto PDL poly-D-lysine, laminin coated plates. After 13 minutes, growth media was added to culture. Media was half-changed 3 to 4 days after initial culture.

## Ethics statement

This study was carried out in strict accordance with the recommendations in the Guide for the Care and Use of Laboratory Animals of the National Institutes of Health. The protocol was approved by the Committee on the Ethics of Animal Experiments by Washington University in St. Louis (Protocol Number: 20–0484; 23–0288). Euthanasia was performed by CO2 overdose and cervical dislocation under anesthesia.

## Nocodazole treatment

At 6 days in vitro (DIV), neurons were pretreated with DMSO or 500nM DLK inhibitor (GNE3511) resuspended in neurobasal (Gibco). After 15 minutes, neurons were then treated with DMSO, or 200nM nocodazole. This produced 4 groups ((1)DMSO, DMSO; (2) DMSO, Noc; (3) DLKi, DMSO and (4) DLKi, Noc). Neurons were collected for further analysis or imaging 16 hours after treatment.

## Western blot

Lysate buffers (60 mM Tri-HCl, pH 6.8; 50% glycerol; 2% SDS; 0.1% bromophenol blue) contain protease cocktail (cOmplete, mini, EDTA-free protease inhibitor; 1183617001, Millipore Sigma) and phosphatase inhibitor cocktail (P0044, Millipore Sigma). Lysates were centrifuged at 4˚ for 5 minutes at 5,000g. Laemelli 6x Reducing buffer (Boston Biosciences) was added at 1x final concentration and boiled for 10 minutes. Samples were run on Mini-PROTEAN® TGX Precast Protein Gels (Bio-Rad Laboratories) and subsequently transferred to nitrocellulose membranes using Trans-Blot® Turbo Midi Nitrocellulose Transfer Packs (Bio-Rad Laboratories) on a Trans-Blot Turbo Transfer System. Membranes were blocked in TBS+0.1% tween and milk or 5% bovine serum albumin (Sigma-Aldrich, cat# A3912-50G) in TBS+0.1% tween (phospho- protein samples) for 60 minutes.

The following antibodies were used: 1:1000 rabbit anti-MKK4 (Cell Signaling Technologies 9152), 1:1000 rabbit anti-phospho-MKK4 (ser257/thr261) (Cell Signaling Technologies 9156),1:1000 rabbit anti-phosphorylated cJun (ser73) (Cell Signaling Technologies 3270S), 1:1000 rabbit anti-cJun (Cell Signaling Technologies 9165P), 1:1000 rabbit anti-HSP90 (Cell Signaling Technologies 4877S). Goat Anti-Rabbit IgG (Jackson ImmunoResearch Laboratories 111–035–045). For phospho-proteins, membranes were stripped with Pierce™ Restore™ PLUS Western Blot Stripping Buffer(Thermo Scientific 46430) for 15 minutes at 37˚, washed 3x 15 minutes and re-processed for non phosphorylated protein. Raw images are available (S1 Raw images).

## Sequencing

After nocodazole treatment, RNA was extracted from eDRGS. For each sample, 3 wells of a 12 well plate (corning) were polled. RNA extraction was performed using the ReliaPrep RNA Cell Miniprep System (Promega). All biological replicates were performed on the same day. For each condition, 4 replicates were obtained. RNA sequencing data available at NCBI GEO: GSE244883.

**Table 1. qPCR primers used in this study.**

| Target | interest | efficiency | Forward Primer | Reverse Primer |
|--------|----------|------------|----------------|----------------|
| ubc | housekeeping | 97 | GCCCAGTGTTACCACCAAGA | CCCATCACACCCAAGAACA |
| rpl27 | housekeeping | 108 | AAGCCGTCATCGTGAAGAACA | CTTGATCTTGGATCGCTTGGC |
| zyx | actin | 112.7 | AGCGAGCCTCCTGTAGC | GGGCAAAGGAAAGTCTTCCG |
| itga9 | actin | 100 | TCAAGTGCAGTGTGGGATTT | CAGTCACGATGAAGCTGAGAA |
| stmn4 | cyto | 90 | GAACAGCTCGCTTTGGAAAC | CTGGAATCTGGTGGCTGAAT |

## Data processing

Reads were processed using the nf-core/rnaseq workflow version 3.6 using STAR and Salmon for alignment and quantification and default settings. QC metrics from the nf-core pipeline, collected by multiqc, were inspected visually for evidence of outliers (see supplement). None were found. Differential expression analysis was performed using DESeq2 version 1.34.0 with an experiment design of ~researcher + pretreatment * treatment. The ashr method of shrinkage was applied to the coefficients. The code used to execute DESeq2, extract the results, and perform the paper analysis is available at (https://github.com/Chase-Mateusiak/DLK_rnaseq).

## qPCR

Biological replicates were obtained on independent days. RNA was collected and used to create cDNA library (Qscript, cDNA Supermix, Quanta bio). For each reaction, 80 ng cDNA was used. Each biological experiment was performed in triplicate. Primers used for qPCR are presented in Table 1. Obvious outliers in technical replicates were eliminated from further analysis. Technical replicates were averaged to create a biological replicate. Biological replicates with greater than 0.5 SD were eliminated from further analysis. Pfaff method was used for quantification. Primer efficiency was calculated using pooled cDNA and 4x, 10 fold dilution series in an independent experiment.

## GO enrichment

In the DLKi in Noc data set, we ranked genes by log2FoldChange and selected the 500 greatest absolute value changes. This set was analyzed using gene set enrichment analysis with WebGestalt. Additional GO enrichment analysis was performed using gProfiler, using differentially expressed genes with a greater than absolute value log2foldchange greater than 0.5.

## Immunohistochemistry

Neurons were cultured on 4 well slides. At 1DIV, neurons were transduced with GFP lentivirus. On 6DIV, neurons were treated with nocodazole and fixed for 10 minutes in 2%PFA. Samples were washed 3x 5 minutes and subsequently stained with Phalloidin, CF568 Conjugate (Biotium) for 30 minutes. Neurons stained for Tuj1 (T2200, Sigma) 1:1000. Neurons were washed 2x 5 minutes and mounted using VECTASHIELD® HardSet™ Antifade Mounting Medium, Liquid (Vector Laboratories).

To investigate microtubule integrity, neurons were briefly washed (1 minute) in cytoskeletal extraction buffer (1 mM Magnesium Chloride, 1mM EGTA, 0.1M 1,4-Piperazinediethanesulfonic acid (PIPES), 0.5% Triton X-100), fixed for 10 minutes in 1% glutaraldehyde (Electron Microscopy Sciences 16019), reduced in freshly prepared 0.1% Sodium Borohydride (Sigma 452882), washed 2x5 in PBS, and stained using the Immunostaining media kit (Active Motif,

cat. no. 15251). Neurons were stained with 1:500 detyrosinated tubulin antibody (Sigma AB3201) and neurofilament 200 (Sigma N4142).

## Quantification of neuronal morphology

To mark neuronal morphology, eDRGs were transduced with GFP lentivirus. FUGW (Addgene 14883)(200ng), vesicular stomatitis virus G (200 ng) and pSPAX2 (600 ng) using FuGENE 6 (Promega) were cotransfected into HEK 293 cells for lentivirus production HEK 293 cells were plated at 60% density the previous day. Lentiviral supernatant was collected after 2 days and applied to eDRG cultured at no greater than 10% volume.

Cell bodies were plated on the edge of 24 well plates (corning), to encourage axon and growth cone density in the center of the plate. At 6DIV, neurons were treated as described above. After 16 hours, neurons were imaged using the 40x lens of the IN Cell Analyzer 6000 (GE). For each well, 60 images were acquired and manual sorted for images with axon tips. These images were cropped to 150 x 150 pixel images. Images were subsequently processed in ImageJ using the following steps 1) Make 8bit. 2) Adjust Local Autocontrast (Phansalkar. Radius 15) 3) Invert image 4) Despeckle 5) Analyze Particles (Total Area, %Area, Particle Size: 5-infinity).

To stain for actin processes, neurons were cultured as described above. Neurons were fixed in 2% PFA for 10 minutes, and washed in PBST 3 x 5 minutes. Neurons were stained with Phalloidin, CF568 Conjugate (Biotium) for 30 minutes. To quantify Tuj1 integrity, neurons were fixed in 2%PFA for 10 minutes, stained with Tuj1 (Rb, 1:1000; MAB1637, Sigma). Neurons were imaged at 20x on the IN Cell Analyzer 6000 (GE). Images were analyzed using the following steps 1) Make 8bit 2) Adjust Local Autocontrast (Phansalkar. Radius 25) 3) Analyze Particles (size 5-$\infty$; circularity 0.5–1) 4) Collect %Area. Measurements were normalized to DMSO controls collected on the same day.

## Statistical analysis

All graphs and statistical analysis was performed in R. Code used to generate graphs and analysis available at: https://github.com/laura-devault/DLK_figures

## Results and discussion

### Treatment with low dose nocodazole induces DLK-dependent signal transduction in cultured neurons

Treatment with low dose nocodazole, a cytoskeletal perturbing drug, increases axon regeneration. This response requires DLK and JNK [10], suggesting that low dose nocodazole treatment activates DLK and downstream MAP kinases [26,27]. To test this facet of our hypothesis, we evaluated the interaction of low dose nocodazole and DLK inhibitor in cultured DRG neurons by pretreatment with DLK inhibitor (500nM) and subsequent 16 hour treatment with low dose nocodazole (200nM). This dose of nocodazole mildly perturbs the cytoskeleton, diminishing microtubule polymerization, but not microtubule density, after 4 hours [28]. While nocodazole concentration may decline over the course of 16 hours, concentration will be consistent within experimental groups. To determine if the DLK signaling cascade is activated in these conditions, we examined the phosphorylation of MKK4, a direct target of DLK [29]. After 4 hours of treatment, there is an increase in the ratio of phosphorylated MKK4 to MKK4, indicating the activation of the MAP3K signaling cascade (Fig 1A and 1B). Previous studies demonstrate that activation of a DLK retrograde signaling cascade induces transcription and phosphorylation of cJun, a transcription factor associated with regenerationhttps://www.zotero.org/google-docs/?sn9UUG [5]. Indeed, treatment with low dose nocodazole was

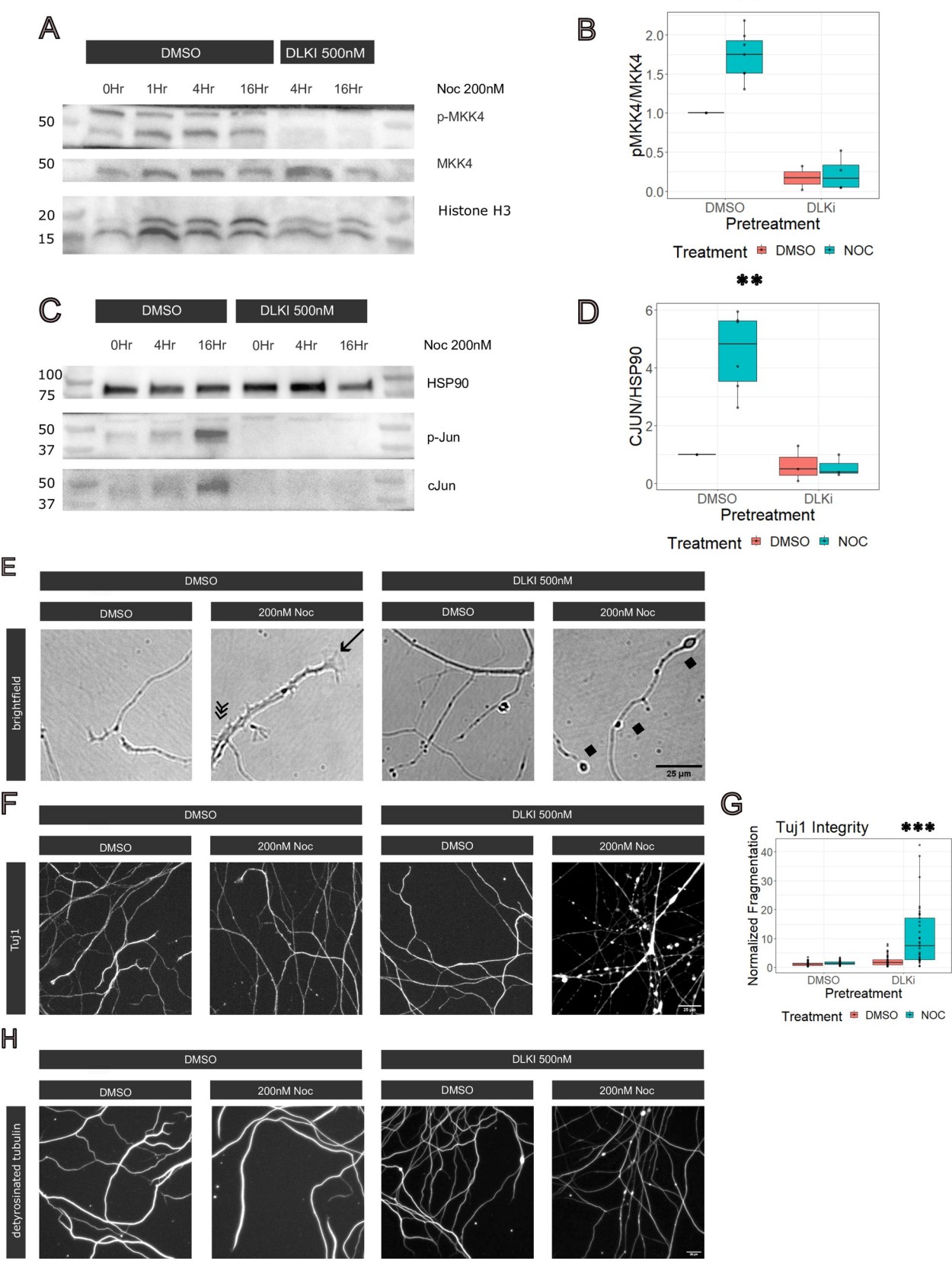

**Fig 1. Treatment with low dose nocodazole activates DLK dependent signaling, resulting in transcriptional changes.** A) Immunoblot for phospho-MKK4, MKK4 and HSP90. Cultured DRGs were pretreated with DMSO or 500nM DLKi (GNE-3511) and subsequently treated with 200nM nocodazole or DMSO. Samples were collected at 4 and 16 Hours. Non-specific band in p-MKK4 blot is routinely noted in preparations of DRGs. B) Quantification of immunoblot. A two-way ANOVA revealed there was a statistically significant interaction on the ratio of pMKK4/MKK4 between the effect of DLKi and low dose nocodazole treatment (Df = 16, F = 9.329, p = 0.008). C) Immunoblot for phospho-Jun, cJun and HSP90. D) Quantification of immunoblot. A two-way ANOVA revealed there was a statistically significant interaction expression of cJun between the effect of DLKi and low dose nocodazole treatment (Df = 14, F = 17.04, p = 0.001). E) Representative images of cultured neuron axons after 16 hours of treatment. Lamellipodia structures are noted with an arrow; short branching structures are noted with a triple arrow and axonal swellings are noted with diamonds. F) Axons stained for Tuj1 after 16 hours of treatment. G) Normalized microtubule fragmentation after 16 hours of treatment. A two-way ANOVA revealed a statistically significant interaction between the effect of DLKi and low dose nocodazole treatment (Df = 153, F = 24.16, p = 2.26 x 10–6).

previously shown to induce cJun phosphorylation in DRGs in a DLK dependent manner [10]. We verified that low dose nocodazole induced cJun protein expression in a DLK-dependent manner (Fig 1C and 1D). These results confirm that treatment with low dose nocodazole activates DLK signaling resulting in increased expression of the transcription factor cJun, thus demonstrating criteria (1) cytoskeletal perturbation activates the DLK signaling cascade.

We examined neuronal cultures to determine the effect of nocodazole on axon morphology. After 16 hours of nocodazole treatment, we consistently observed prominent growth cones consisting of broad lamellipodial structures as well as numerous short branches extending from the primary axon (Fig 1E). These nocodazole-dependent effects did not occur with DLKi pretreatment. In addition, pretreatment with DLKi inhibitor independently changed axon morphology. Within minutes of DLK inhibitor application, growth cones disappeared, consistent with the role of activated JNK signaling in axon outgrowth [30]. By 16 hours after pretreatment with DLK inhibitor, swellings and protrusions appeared along the most distal portions of the axon. In subsequent experiments, we characterized and quantified changes to axonal morphology.

Since nocodazole inhibits microtubule polymerization, we examined the impact on microtubule integrity. As predicted from prior studies which use low dose of nocodazole [28], microtubule integrity was maintained in control and nocodazole treated neurons. To test for changes in microtubule distribution, we stained for the neuronal tubulin marker, Tubb3, Tuj1 (Fig 1F and 1G). Loss of Tubb3 has previously been shown to lead to microtubule regrowth defects in DRGs *in vivo* [31,32]. To determine the DLK dependence of Tubb3 distribution, we evaluated the effect of DLK inhibitor on neurons treated with DMSO (control) and nocodazole. We observed no defects in DLK inhibitor treated neurons. However, DLK inhibition in nocodazole-treated neurons causes defects in Tuj1 staining including gapping within the microtubules leading to a bead-on-a-string appearance along the axon. Quantification of microtubule continuity demonstrates a significant defect upon DLK inhibition in nocodazole-treated neurons (Fig 1F and 1G). This suggests that DLK signaling is required to maintain Tubb3 across axons, potentially facilitating microtubule regrowth and repair. To complement these experiments, we asked if the distribution of stable microtubule populations changed upon DLK signaling perturbation using a microtubule preservation protocol [33] to stain for detyrosinated microtubule populations. We observed continuous detyrosinated tubulin staining across all conditions. DLK inhibition in nocodazole-treated neurons led to circular microtubule structures after treatment (Fig 1H), suggesting some abnormalities, but not the frank fragmentation suggested by the Tuj1 staining. We next investigated if transcriptional changes could contribute to these alterations in axon morphology.

## Transcriptional profiling reveals a DLK-dependent genetic program activated by low dose nocodazole treatment

To gain insights into DLK-dependent transcriptional changes following cytoskeletal insult, we performed bulk RNA sequencing on cultured neurons treated for 16 hours with low dose

nocodazole. Using a two-factor completely crossed experimental design [34], we determined the transcriptional effects of nocodazole treatment, DLK inhibition and the interaction of nocodazole and DLK inhibitor (S1 File). Our analysis identified DLK-dependent effects of nocodazole (Fig 2A). In a principal components analysis, we observed that treatment with nocodazole led to a DLK-dependent clustering of samples along PC1, an axis accounting for 57% of the variability in the transcriptome (Fig 2B). This suggests that cytoskeletal perturbation has a major effect on neurons' transcriptional profiles (corresponding to a major shift on PC1) that is abrogated by DLK inhibitor.

Among the DLK-dependent nocodazole-dependent differentially expressed genes, the four most upregulated genes were the transcription factors Jun, Egr1, and Atf3 and the signaling regulator, Dusp1 (Fig 2C). Indeed, increased Jun transcription is consistent with the increase in its protein expression which we observed after nocodazole treatment (Fig 1C and 1D). Jun is a downstream target of the DLK/JNK pathway, which is phosphorylated and activated by JNK. Hence, the transcriptional upregulation of Jun is likely an amplification of the DLK signal, which can lead to apoptosis if the induction is sustained [35]. Concurrently, upregulation of Dual Specificity Protein Phosphatase (DUSP) genes indicates the activation of pathways that attenuate the same DLK signal. This suggests a feedback mechanism which is a hallmark of a homeostatic system. DUSPs are potent negative regulators of MAP kinase activity. DUSPs main function is to dephosphorylate and inactivate MAPKs [36]. Expression of both cytoplasmic DUSPs associated with the p38/JNK pathway (Dusp16, Dusp8, and Dusp10) and inducible nuclear DUSPs (Dusp1, Dusp4, and Dusp5) increased after nocodazole treatment in a DLK-dependent manner. Additional negative phosphorylation regulators (Timp3, Cnksr3, Trib1, Per1, Spry1, Sfrp4, Spred2, Ppp1r15a, Spred3, Sema6a, Inpp5j, Lrrk2, Casp3) are differentially expressed after nocodazole treatment in a DLK-dependent manner (Fig 2E). Previous work demonstrates that serine-threonine phosphatases, including PP2A and Ste20 Kinases [37,38], impact DLK signaling. We searched for differentially regulated serine-threonine kinases, but found few significant DLK-dependent changes after nocodazole treatment other than modest changes in the expression levels of the serine-threonine phosphatases Ppp4r4, Ppp1r15a and Ptprr (S1 File). Taken together, these finding demonstrate that, nocodazole treatment induces DLK-dependent transcriptional changes that likely regulate and attenuate DLK signaling, thereby supporting homeostatic criteria (3) that a cytoskeletal homeostatic program attenuates signaling and returns the cell to baseline.

To gain a more comprehensive understanding of the differentially expressed genes that are both DLK-dependent and nocodazole-dependent, we used Gene Ontology (GO) term based gene set enrichment analysis (GSEA) (Fig 2D–2I). Many of the downregulated GO terms are associated with the function of mature neurons. These include terms associated with synapses, neurotransmitter, neuropeptides, behavior, pain, and motor function. In contrast, the upregulated terms are generally associated with stressed or immature/regenerating neurons, kinase regulation, and cytoskeleton, including terms for polarized growth, response to ER stress, kinase regulation, transcriptional regulation, and the actin-rich cell cortex and microtubule-rich centrosomes. Taken together, this suggests that nocodazole treatment activates a DLK-dependent program that regulates transcriptional activity through transcription factors and MAP kinase regulators. Activation of this program downregulates mature neuron behaviors and upregulates both stress response and growth genes. This program increases expression of genes related to polarized neuron growth, including genes regulating the cytoskeleton.

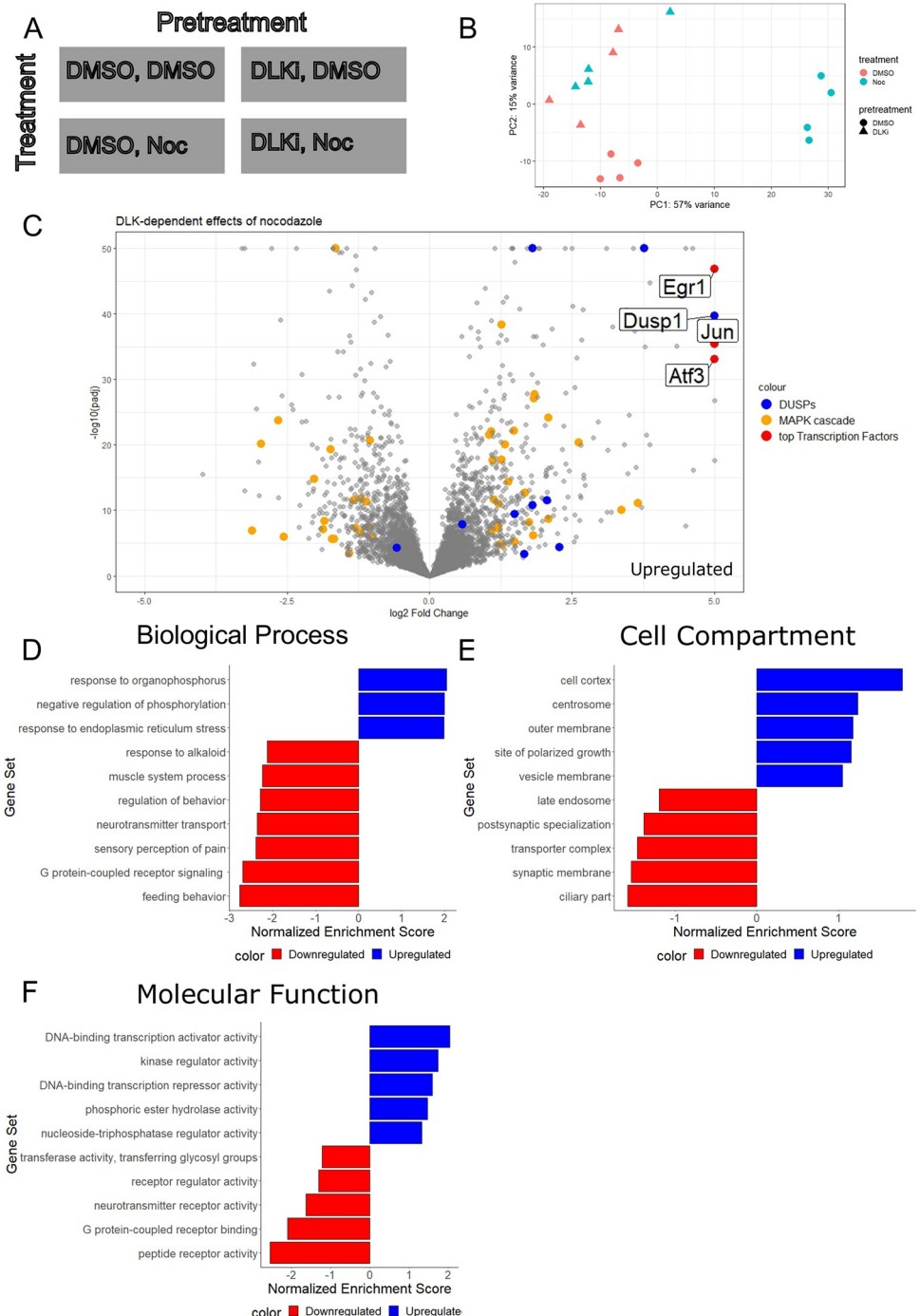

**Fig 2. DLK inhibition on low dose nocodazole treatment impacts gene expression. A) Experimental design for RNA sequencing with two factors, pretreatment (DMSO, DLKi) and treatment (DMSO, Noc).** B) Principal Component Analysis (PCA) of RNA sequencing experience for four conditions (DMSO, DMSO; DMSO, Noc; DLKi, DMSO; DLKi, Noc) with four replicates. C) Differential gene expression in cultured DRGs of DLKi in low dose nocodazole. Log2FoldChange was determined in a two factor with interaction design in DESeqEQ2. Log2FoldChange was plotted with regard to base mean gene expression. Large changes in transcription factors (Egr1, Jun and Atf3), DUSP and MAPK cascade genes are noted in red, blue and yellow. MAPK cascade (GO:0000165) is significantly enriched in genes with a greater than 1 log2FoldChange (p = 1.6 x 10–9). Genes with the greatest log2FoldChange (n = 500) were selected for further analysis using WebGestalt for Gene Set Enrichment. D) Biological Process enrichment in response to ER stress (value) and negative regulation of phosphorylation (value). Nocodazole treatment increases gene expression related to Stress Response and Signaling Regulation (ER stress (GO:0034976, NES = -1.9892,

p = 0.00147, FDR = 0.09715) and Negative Regulation of Phosphorylation (GO:0042326, NES = -1.9969, p = 0.0037406, FDR = 0.12066)). Conversely, nocodazole treatment decreases expression of biological processes specific to mature neuronal behaviors including Pain Perception, Behavior Regulation and Feeding Behavior (Sensory Perception of Pain (GO:0019233, NES = 2.3965, p = <2.2e-16, FDR = 0.008895), Regulation of Behavior (GO:0050795, NES = 2.29, p = <2.2e-16, FDR = 0.01129), Neurotransmitter Transport (GO:0006836, NES = 2.3623, p = <2.2e-16, FDR = 0.00975), G-protein Coupled Receptor Signaling (GO:0007187, NES = 2.6953, p = <2.2e-16, FDR = <2.2e-16) and Feeding Behavior (GO:0007631, NES = 2.772, p<2.2e-16, FDR<2.2e-16)). E) Cell compartment enrichment in cell cortex (GO:0005938, FDR 0.28243, p = 0.0057389), neuron projection terminus (GO:0044306, FDR 0.46809, p = 0.006) F) Molecular function enrichment in DNA-binding Transcription Activator Activity increased after nocodazole treatment in a DLK-dependent manner (NES: -2.0360, FDR: 0.043388, GO:0001228, p = 0.003). Conversely, Peptide Receptor, Neurotransmitter, and G-protein Coupled Receptor Activity (peptide receptor activity (NES: 2.5456, FDR: 0.00084809, GO:0001653, p < 2.2e-16), and G protein-Coupled Receptor Binding (NES: 2.0976, FDR: 0.024595, GO:0001664, p = 0.0056338) had positive enrichment scores.

## Low dose nocodazole induces DLK-dependent transcriptional and morphological changes to the actin cytoskeleton

We noted that GO and KEGG pathway terms related to the cytoskeleton were significantly overrepresented in our data set. KEGG pathway analysis identified significant enrichment in Focal Adhesions, driven by differential expression of Jun, Itga5, Shc4, Zyx, Shc3, Tnc, Itga9, Bcl2, Pxn, Flnb, Pik3cb (Fig 3A). Focal adhesions are associated with actin-integrin structures which regulate actin dynamics. In addition, GO GSEA of differentially expressed genes revealed enrichment in the Actin Cytoskeleton and Cytoskeleton Organization (Fig 3B). Differentially expressed genes with at least a four-fold change included 12 genes associated with Cytoskeletal Organization, of which 9 belong to Actin Cytoskeleton Organization including: Sipa1l1, Zyx, Rnd3, Palld, Phldb2, Rhpn2, Ptger4, Tmod1, Bcl6 and Arc. Arc regulates actin in dendritic spine formation [39]. Lrguk is associated with ciliary assembly [40], while Stmn4 is a well-established microtubule regulator that sequesters free tubulin [41]. Stmn4 is notable as a highly expressed transcript consistently associated with DLK data sets (Fig 3B) [42–44]. These findings suggest that nocodazole treatment induces a transcriptional program that affects the cytoskeleton primarily through actin regulation, fulfilling criteria (4) that a cytoskeletal homeostatic program upregulates expression of cytoskeletal genes. We independently verified a subset of these differentially expressed genes by quantitative PCR (qPCR) (Fig 3C).

To investigate whether morphological alterations to the actin cytoskeleton occurred in concert with the transcriptional changes, we examined the axon and growth cone. Along the axon, short protrusions emerge after nocodazole treatment (Fig 4A). These protrusions resemble actin-rich filopodia and immunostaining with the f-actin marker phalloidin confirmed the presence of actin so we identify these protrusions as filopodia [45]. The number of filopodia significantly increased after 16 hours of nocodazole treatment (Fig 4C). In addition to regulating filopodia, increased actin dynamics at the growth cone also mediates axon extension. Therefore, we tested the influence of nocodazole on growth cone morphology, and observed a significant increase in the size of the growth cone following nocodazole treatment (Fig 4B and 4D). Both the increased number of filopodia and larger size of growth cones is DLK dependent (Fig 4A–4D). These findings demonstrate that cytoskeletal perturbation triggers DLK-dependent changes in actin morphology, addressing criteria (5) that cytoskeletal repair occurs in a DLK-dependent manner.

Taken together, these findings demonstrate that low-dose nocodazole treatment induces DLK-dependent changes to the actin cytoskeleton, with upregulation of genes associated with Actin Cytoskeleton Organization and concomitant emergence of actin-rich filopodia along the axon and an increase in size of the growth cone. Hence, disrupting the cytoskeleton via low dose nocodazole leads to DLK-dependent transcriptional and morphological changes to the neuronal actin cytoskeleton.

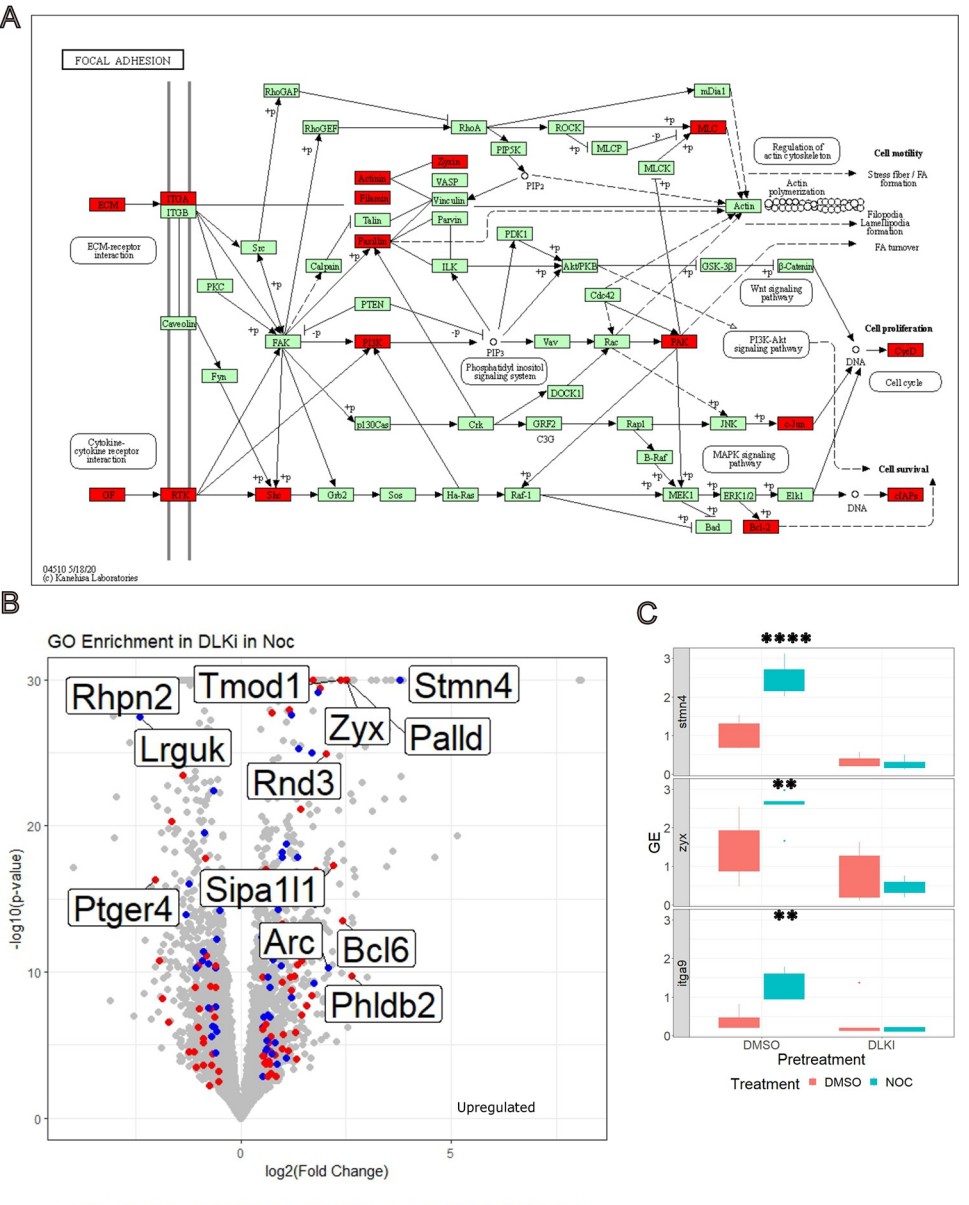

**Fig 3. Treatment with low dose nocodazole increases expression of cytoskeleton organization genes.** A) KEGG diagram for focal adhesion NES = -2.4 (mmu04510, FDR = 0.0026919, p = <2.2e-16). Genes with a log2FoldChange > -0.5 are noted in red. B) Differential gene expression of DLK inhibitor in low dose nocodazole in cultured DRGs, actin cytoskeleton organization (GO:0030036, padj = $1.979 \times 10^{-7}$) noted in red, and cytoskeleton organization (GO:0007010, padj = $5.452 \times 10^{-5}$) noted in blue. Cytoskeletal organization genes with a log2FoldChange greater than 2 noted. C) Differential expression of select genes (Stmn4, Zyx, Itga9) was confirmed by qPCR. Expression was normalized to Reference Genes (Ubc, Rpl27). There was a statistically significant interaction between low dose nocodazole treatment and DLK inhibitor on gene expression for Stmn4 (F = 19.45, p = 0.000851), Zyx (F = 4.580, p = 0.05190) and itga9 (F = 9.053, p = 0.0101).

## DLK-dependent transcriptional changes after nocodazole treatment are similar to transcriptional changes after NGF deprivation

To investigate the existence of a DLK-dependent transcriptional signature common to various insults that cause neuronal cytoskeletal stress, we compared the DLK-dependent effects of

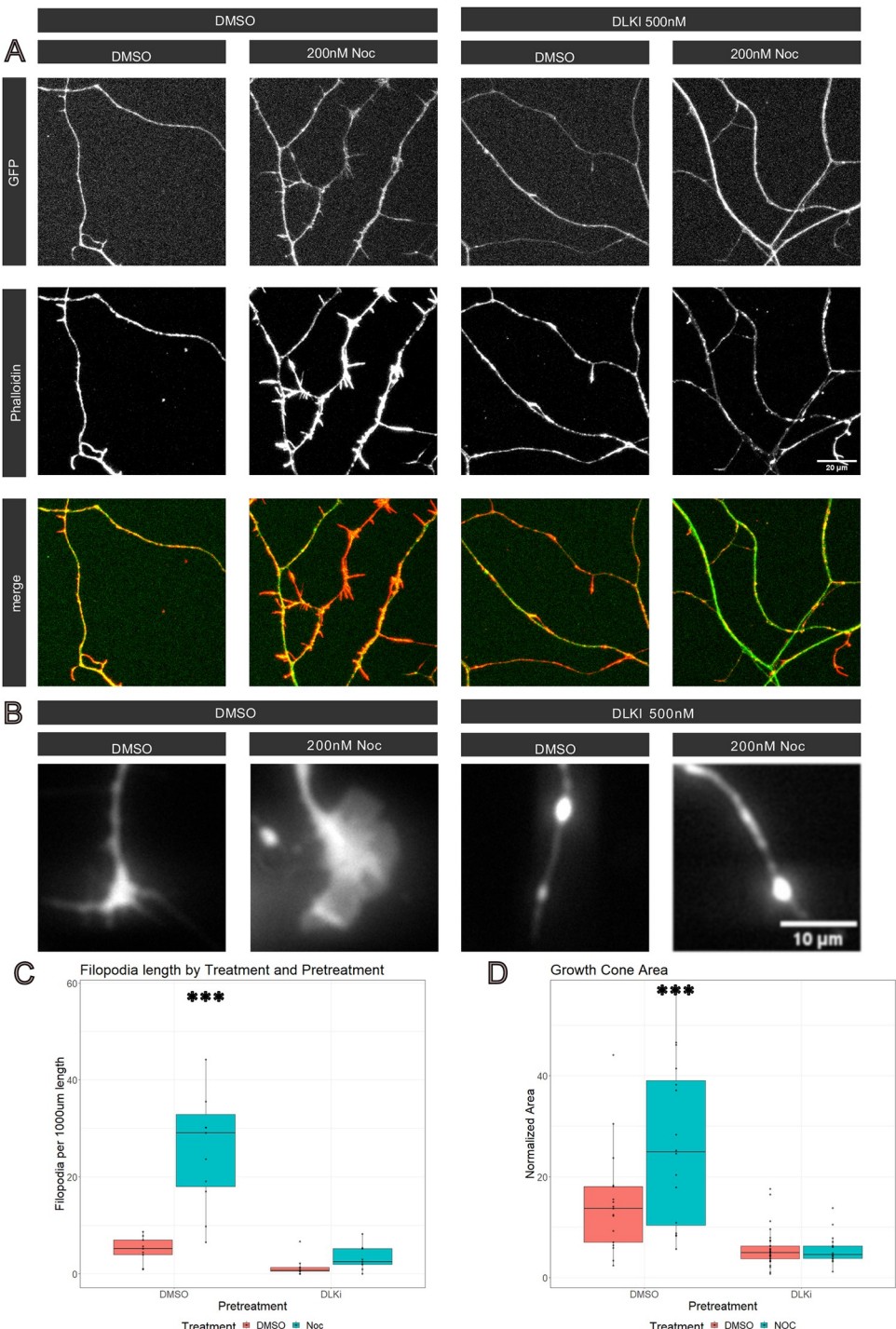

**Fig 4. Low dose nocodazole increased filopodia along the axon in a DLK-dependent manner.** A) Cultured DRG neurons axons expressing GFP were stained for phalloidin after 16 hours of treatment with either low dose nocodazole or DMSO (control). Neurons were pretreated with either DLKi or DMSO. B) Differences in growth cone morphology were observed at 16 hours after treatment C) We further analyzed the number of filopodia along branches and found a significant increase in short branches (less than 10 μm) in neurons treated with DMSO+Noc (27.5 +/-10.1) compared to DMSO+DMSO treated neurons (4.9 +/- 2.1), DLKi+DMSO (1.4 +/- 1.35) DLKi+Noc (3.2 +/- 2.0). A Two-way ANOVA confirmed that the presence of actin rich branches in low dose nocodazole treated neurons was DLK dependent (Df = 35, F = 15.35, p = 0.0004). D) Normalized Growth Cone area was significantly increased after low dose nocodazole treatment. A two-way ANOVA demonstrated increased growth cone area in Noc (26.5% +/- 8.6%) compared to DMSO treated neurons (14.8% +/- 5.4%), DLKi (5.7% +/- 1.5%) DLKiNoc (5.6% +/- 1.5%). (Df = 76, F = 8.0, p = 0.006).

nocodazole to those induced by NGF deprivation. To do this, we compared our data (effect_of_dlki_noc) to the effects of NGF deprivation versus DLK inhibitor (dlki_vs_dmso_ngf_minus) in cultured DRGs [42].

We found a large and highly significant correlation between differentially expressed genes after nocodazole treatment and NGF deprivation in cultured sensory neurons (Fig 5A). We identified 1098 differentially expressed genes shared between the two data sets (Fig 5B and 5C). Among these, thirteen genes exhibited a greater than four-fold gene expression change in the same direction in both datasets (Dusp1, Htr4, Atf3, Irf6, Dusp16, Ypel4, Arid5a, Egr1, Dact2, Gprc5c, Jun, Creb5, Sh2d3c), demonstrating that MAP kinase negative regulators (Dusp1, Dusp16), transcription factors (Atf3, Egr1, Jun, Creb5, Irf6) transcriptional regulators (Arid5a, Dact2), scaffold proteins (Sh2d3c), membrane integrity regulators (Ypel4) and G-coupled proteins (Gprc5c) are abundantly upregulated by DLK signaling in both paradigms. Only one gene, Htr4, a serotonin receptor, has at least a four-fold downregulation after both NGF deprivation and nocodazole treatment, consistent with a downregulation of more mature neuronal functions. These results suggest that DLK activation, either through low dose nocodazole treatment or NGF deprivation, results in a shared transcriptional signature. Despite this shared transcriptional signature, treatment with low dose nocodazole or NGF deprivation results in different neuronal phenotypes, enhanced growth or cell death, suggesting that the impact of the transcriptional signature is context dependent.

To test the hypothesis that DLK-dependent signaling results in consistent transcriptional changes to actin regulators between the low dose nocodazole and NGF deprivation datasets, we investigated the GO term Actin Cytoskeleton Organization (Fig 5D and 5E). Indeed, there is a significant enrichment of this term in both datasets (Low Dose Noc p = $1.626 \times 10^{-8}$ and NGF Deprivation p = $8.656 \times 10^{-11}$). In both datasets, Zyx, which coordinates actin dynamics at focal adhesions and stress fibers, is significantly upregulated. This perhaps accounts for the morphological changes that we observe in nocodazole treated neurons. We also observe a greater than two-fold downregulation of the following: Fmnl1, a formin-like protein, Tacr1, a transmembrane receptor and Rhpn2, a Rho-GTPase binding protein which may limit stress fiber formation [46]. These results suggest that diverse mechanisms of DLK activation result in differential expression of actin cytoskeleton regulators, consistent with the hypothesis that neurons engage a DLK-dependent cytoskeletal homeostatic network in response to perturbation.

## Conclusions

In this work, we explore the hypothesis that DLK regulates a homeostatic network that responds to cytoskeletal damage and facilitates cytoskeletal repair, a process we term the Cytoskeletal Stress Response (CSR) pathway. To determine if this pathway exists, we evaluate the following five criteria in this paper: (1) cytoskeletal perturbation must initiate a DLK signaling cascade resulting in (2) transcriptional changes which include (3) genes capable of attenuating the strength of the DLK signaling cascade and (4) genes capable of modulating the cytoskeleton, thus resulting in (5) changes to the cytoskeleton. Here, we discuss the current evidence for a CSR-pathway and identify existing gaps in evidence required to establish the CSR as a homeostatic response.

### DLK plays a critical role in sensing and responding to cytoskeletal damage

We demonstrate that cytoskeletal damage activates a DLK signaling cascade. This result is consistent with published work identifying pharmacologic and genetic perturbations of the cytoskeleton that activate DLK dependent stress signaling in neurons [3,5,10,15,16,18,47]. While

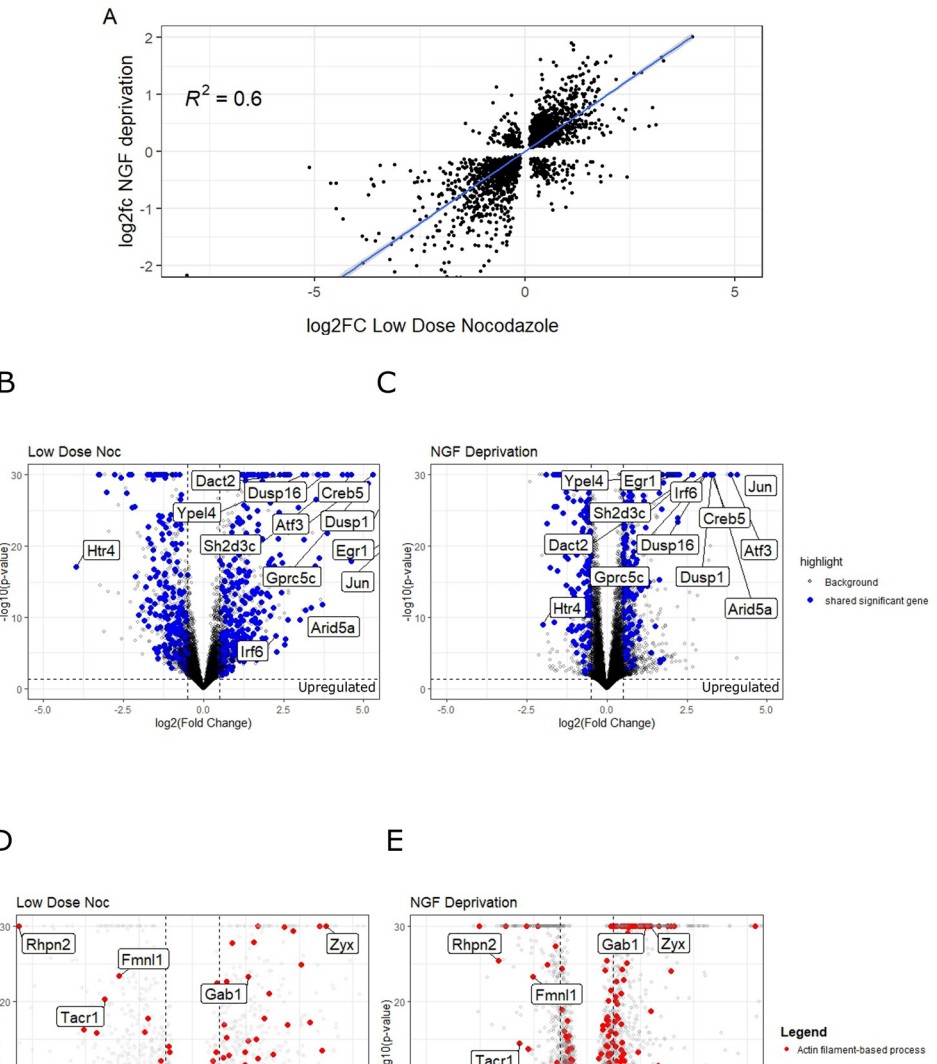

**Fig 5. Conservation of DLK-dependent injury gene expression after low dose nocodazole treatment and NGF deprivation.** Linear models were used to analyze the similarities between the different treatments. Correlation between log2FoldChange of differential gene expression of DRGs with respect to the effect of DLKi in low dose nocodazole and (A) log2FoldChange of DLKi vs NGF deprivation (correlation p-value < 1.2 e-12). (B,C) Transcriptional comparison of differential expression in Low Dose Nocodazole and NGF deprivation paradigms, genes differentially expressed in both data sets marked in blue. Volcano plot of -log10(p-adjusted) v log2Fold Change for (C) Low Dose Nocodazole and (D) NGF Deprivation. Genes differentially expressed in both data sets with greater than 2 log2FoldChange are noted. (D,E) Actin Filament based processes (GO:0030029) are enriched in differentially expressed genes sets for Low Dose Noc (p = $1.626 \times 10^{-8}$) and NGF Deprivation (p = $8.656 \times 10^{-11}$). Genes belonging to GO: Actin filament-based process (GO:0030029) displayed in red. Actin Cytoskeleton Organization genes (GO:0030029) with absolute log2Fold Change > 1 present in both datasets are noted (Zyx, Gab1, Fmnl1, Tacr1, Rhpn2).

these findings support the idea that cytoskeletal stress activates DLK, we lack a mechanistic understanding of how cytoskeletal disruption activates DLK. In fact, seemingly disparate genetic perturbations of the cytoskeleton activate DLK signaling, including mutations in

Spectraplakin, an actin microtubule linking protein, Camsap, a minus end microtubule capping protein, and Beta-spectrin, a scaffolding protein essential for membrane integrity [48]. The exception to this set of unknown intermediaries between perturbation and DLK activation is Rae1, a WD40 microtubule interactor [49,50], which affects DLK via regulating the levels of the E3 ubiquitin-protein ligase, Highwire [17], which in turn ubiquitinates and promotes the degradation of DLK [41,51]. In sum, the mechanisms linking cytoskeletal perturbation to DLK activation are almost entirely unknown.

Unlike the intermediaries between cytoskeletal perturbation and DLK activation, the DLK signaling cascade itself is well established. DLK phosphorylates a MAP2K, either MKK4 or MKK7, which in turn phosphorylates a MAPK, either JNK or p38 [20,24]. Phosphorylation and palmitoylation of DLK and JNK drives association with retrograde trafficked vesicles [52,53]. The retrograde trafficking of these signaling molecules results in increased transcription and phosphorylation of the transcription factor, Jun, which is critical for DLK-dependent increases in neuron regeneration [7,22]. We find that the transcription factors, Jun, Egr1, and Atf3 exhibit the greatest DLK-dependent changes after cytoskeletal disruption. While Jun can promote axon regeneration, it is also a key driver of apoptosis [54], and DLK-dependent retrograde signaling promotes retinal ganglion cell death after optic nerve injury [24,55]. Interestingly, the DLK-dependent expression of DUSPs, which negatively regulate DLK/JNK signaling, and so likely mitigate the strength of the Jun signal, may play a key role in ensuring that DLK activation results in axon regeneration rather than apoptosis in DRG neurons. Furthermore, DUSPs may act upon the cytoskeleton. Of particular interest, Dusp1 regulates axonal branching through JNK-dependent cytoskeletal remodeling [56]. Interestingly, Atf3 promotes axon regeneration in retinal ganglion cell models [57,58], but can also regulate degenerative pathways [59]. Egr1, an immediate early gene associated with neuronal activity, is also linked to regeneration in spinal cord injury [60,61]. It will be important to understand the role of Atf3, Egr1 and the DUSP genes in the DLK-dependent axon injury program.

Evidence for homeostatic repair comes from observations of axon growth and maintenance of cytoskeletal integrity. We demonstrate that DLK is required for alteration of the axonal actin and microtubule cytoskeleton after perturbation with nocodazole. While we did not prove that this effect is due to the expression of DLK-dependent genes, this finding is consistent with the model that a DLK-dependent mechanism alters the cytoskeleton. Increased axon growth is a well-documented consequence of activating DLK signaling cascades [5,7,8,10,24]. Both phenomena may be mechanistically linked to the genetic changes we observe. We demonstrate that nocodazole treatment induces DLK-dependent expression of actin regulatory genes and results in broad changes to the actin cytoskeleton, including increased filopodia number along the axon and enlarged growth cones. As nocodazole predominantly acts on the microtubule cytoskeleton, the extensive engagement of actin regulatory mechanisms is unexpected. There are two potential explanations for this incongruity. First, our response may reflect coordination between the actin and microtubule cytoskeleton. Initial reorganization of the actin cortical cytoskeleton precedes coordinated polymerization between the actin and microtubule cytoskeleton into primary axon branches [62,63]. As such, our results may reflect an early stage of outgrowth, preceding microtubule extension into the growth cone. Second, the predominance of actin-related transcripts may reflect a unique role for actin regulation in repairing neuronal structures, as suggested by work which demonstrates that actin turnover drives regeneration [64]. Actin structures may also influence microtubule integrity, as microtubule lattice repair occurs more frequently in stress fiber adjacent regions [65]. Although the transcriptional response predominantly consists of actin regulators, transcription of a microtubule regulatory gene, Stmn4, is strongly increased after nocodazole treatment. Future work will explore the importance of Stmn4 for microtubule integrity and extension into the growth cone.

While not demonstrated in this paper, we predict that the CSR repairs the axonal cytoskeleton at the subcellular scale. In particular, it would be interesting if DLK activation promoted repair of the membrane periodic structure of the axon, an interconnected sub membrane structure of repeating β-spectrin and actin rings that provides strength and flexibility to axons [66,67]. Of note, disassembly of the membrane periodic structure (MPS) likely drives DLK activation [14]. Both loss of β-spectrin and NGF deprivation disrupt the MPS and activate DLK signaling [5,13,14]. Moreover, DLK activation is sensitive to the disassembly of the actin MPS after NGF deprivation. This is demonstrated by pretreatment with jasplakinolide, which stabilizes the MPS and blocks the DLK signaling cascade [14]. Finally, cytoskeletal targeting drugs, including latrunculin A disrupt the MPS and activate DLK signaling [14,68,69]. Less is known about the consequences of activating DLK signaling for the MPS and additional studies will be required to understand if activating the CSR facilitates MPS repair and to demonstrate convincingly that cytoskeletal damage engages a DLK-dependent homeostatic network.

## The CSR resembles other homeostatic networks

The CSR mirrors other homeostatic network responses, such as the UPR or heat shock response, in three ways. First, in the proposed CSR, cellular stress initiates, through DLK, a kinase signaling cascade. This is reminiscent of the way IRE acts as a kinase sensor initiating the UPR. Second, homeostatic networks are associated with transcriptional programs driven by transcription factors. After UPR activation, expression of transcription factors, Atf4, Atf6 and XBP1 drive expression of homeostatic effector genes related to protein quality control, lipid synthesis, ER-associated degradation, apoptosis and autophagy. There is strong evidence from this study and others that activated DLK signaling induces transcription through Jun, Egr1 and Atf3. We speculate that these transcription factors control expression of actin regulatory genes, negative JNK signaling regulators and the dampening of genes related to mature neuron function. Future work will delineate the relationship between these transcription factors and their targets in this pathway. Finally, like other homeostatic pathways, activation of the CSR mediates repair or cell death in a context dependent manner. In the CSR, this is particularly evident when comparing the response of retinal ganglion cells in the CNS to dorsal root ganglion cells in the PNS. While DLK activation of DRGs leads to regeneration, DLK activation in RGCs leads to apoptosis [7,12,24,55].

In summary, we propose that DLK acts as a sensor for the Cytoskeletal Stress Response, enabling neurons to sense and repair damage to the cytoskeleton. This model of a DLK-dependent cytoskeletal homeostatic repair mechanism is consistent with over twenty years of data placing DLK at the center of neuronal degeneration and regeneration programs. Here, we contextualize this argument in light of previous research, demonstrate that DLK signaling is required to maintain microtubule structure following disruption of microtubule polymerization, provide transcriptional analysis of DLK dependent gene expression after cytoskeletal perturbation and present evidence of actin cytoskeletal modulation. If validated, this model adds the cytoskeleton to the array of cell biological structures and processes maintained via homeostatic regulatory networks.

## Supporting information

**S1 Raw images. This file contains raw images of Western blots from Fig 1.**
(PDF)

**S1 Data set. This file contains data collected during the course of this paper, which was used to generate Figs 1–5.**
(XLSX)

**S1 File. This file contains results of the two-factor fully crossed experiment testing the effect of nocodazole and DLK inhibitor on transcription.** Tabs contain the transcriptional effect of (1) nocodazole, (2) DLK inhibitor and (3) of DLKi_on_nocodazole.
(XLSX)

**S1 Fig. This supplemental figure contains QC information about the RNA seq data contained in this paper.**
(PDF)

## Acknowledgments

We thank EJ Brace and Joseph Bloom for critical reading of the manuscript, EJ Brace and Margaret Hayne for technical help and members of the DiAntonio and Milbrandt labs for discussion.

## Author Contributions

**Conceptualization:** Laura DeVault, Aaron DiAntonio.

**Data curation:** Laura DeVault, Chase Mateusiak.

**Formal analysis:** Laura DeVault, Chase Mateusiak.

**Funding acquisition:** Laura DeVault, Aaron DiAntonio.

**Investigation:** Laura DeVault, John Palucki, Michael Brent.

**Methodology:** Laura DeVault, Chase Mateusiak, Michael Brent, Aaron DiAntonio.

**Supervision:** Jeffrey Milbrandt, Aaron DiAntonio.

**Visualization:** Laura DeVault, Chase Mateusiak.

**Writing – original draft:** Laura DeVault, Chase Mateusiak, Aaron DiAntonio.

**Writing – review & editing:** Laura DeVault.

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
