## [Decision Letter · Decision Letter 0]

13 Dec 2023

PONE-D-23-37022The response of Dual-Leucine Zipper Kinase (DLK) to nocodazole: evidence for a homeostatic cytoskeletal repair mechanismPLOS ONE

Dear Dr. DiAntonio,

Thank you for submitting your manuscript to PLOS ONE. After careful consideration, we feel that it has merit but does not fully meet PLOS ONE’s publication criteria as it currently stands. Therefore, we invite you to submit a revised version of the manuscript that addresses the points raised during the review process. Reviewer 1 has substantial concerns about the data quality and presentation. The reviewer finds that at times, it is unclear if the conclusions the authors draw are supported by the data. This is an important concern that requires a detailed response, particularly in view of the fact that the reviewer apparently was unable to see the figure legends. The points raised by Reviewer 2 seem more easily to be addressed.

Please submit your revised manuscript within Jan 27 2024 11:59PM. If you will need more time than this to complete your revisions, please reply to this message or contact the journal office at plosone@plos.org. Please include the following items when submitting your revised manuscript:A rebuttal letter that responds to each point raised by the academic editor and reviewer(s). You should upload this letter as a separate file labeled 'Response to Reviewers'.A marked-up copy of your manuscript that highlights changes made to the original version. You should upload this as a separate file labeled 'Revised Manuscript with Track Changes'.An unmarked version of your revised paper without tracked changes. You should upload this as a separate file labeled 'Manuscript'.

We look forward to receiving your revised manuscript.

Kind regards,

Gerhard Wiche, Ph.D.

Academic Editor

PLOS ONE

Journal Requirements:

2. Thank you for stating the following financial disclosure: "LD: F32NS117784 (national institute of neurological disorders and stroke) AD, JM: R01NS087632 (national institute of neurological disorders and stroke) AD: R37NS065053 (national institute of neurological disorders and stroke) MB: GM 141012 (national institute general medical sciences)".

3. Thank you for stating the following in the Acknowledgments Section of your manuscript: "We thank EJ Brace and Joseph Bloom for critical reading of the manuscript, Margaret Hayne for technical help and members of the DiAntonio and Milbrandt labs for discussion. This work was supported by National Institutes of Health grants (F32NS117784 to LD, R01NS087632 to A.D. and J.M., R37NS065053 to A.D. and GM141012 to MB)".

Please remove any funding-related text from the manuscript and let us know how you would like to update your Funding Statement. Currently, your Funding Statement reads as follows: "LD: F32NS117784 (national institute of neurological disorders and stroke) AD, JM: R01NS087632 (national institute of neurological disorders and stroke) AD: R37NS065053 (national institute of neurological disorders and stroke) MB: GM 141012 (national institute general medical sciences)"

4. Thank you for stating the following in the Competing Interests section: "I have read the journal's policy and the authors of this manuscript have the following competing interests".

Reviewers' comments:

Reviewer's Responses to Questions

**Comments to the Author**

1. Is the manuscript technically sound, and do the data support the conclusions?

Reviewer #1: No

Reviewer #2: Partly

2. Has the statistical analysis been performed appropriately and rigorously? 

Reviewer #1: I Don't Know

Reviewer #2: Yes

3. Have the authors made all data underlying the findings in their manuscript fully available?

Reviewer #1: No

Reviewer #2: Yes

4. Is the manuscript presented in an intelligible fashion and written in standard English?

Reviewer #1: Yes

Reviewer #2: Yes

5. Review Comments to the Author

Reviewer #1: DeVault and colleagues aim to define a cytoskeletal stress response pathway that mediates homeostatic responses to axon injury. DLK, a prominent player in axon regeneration and axon degeneration, is proposed to be a critical player in this homeostatic network that senses cytoskeletal injury. Studies rely upon pharmacological manipulation of microtubules using low level nocodazole and a DLK inhibitor. Using these manipulations the authors examine how neuronal transcriptional profiles are altered and aim to identify transcriptional changes regulated by DLK. The reviewer thinks these are interesting and important concepts, as it is unclear how DLK activation and signaling fit into a larger transcriptional program that is affected by cytoskeletal instability. While this is a potentially interesting study, there are substantial concerns about data quality and presentation. At times, it is unclear if the conclusions the authors draw are supported by the data. Below please find detailed comments.

Major comments:

1) The loading of MKK-4 in Figure 1A is reduced in sample treated with both nocodazole and DLK inhibitor. Are effects of DLK inhibitor on phospho-MKK4 in nocodazole due to reduced loading? Reviewer is not sure this level of variability is reflected in the error bars in Figure 4D (p-MKK4 quantitative data). It appears to be almost 50-60 lower MKK4 levels compared to sample with DLK inhibitor alone in Figure 1A blot. Reduces reviewer confidence in this dataset. Please note that Figure 1 has quantitative results for MKK-4 in panel 1D not 1B.

2) Is Jun completely absent with DLK inhibitor application? This appears to be the case in the blots shown in Figure 1C. How does one differentiate effects of DLK inhibition on Jun activation (p-Jun) when Jun can’t be visualized or isn’t detected? Concerns about quantitative data since Jun levels are so low with Dlk inhibitor. How are Jun levels normalized and quantified relative to HSP90 if Jun is not an observable band?

Is the concentration of DLK inhibitor too high? Would it be valuable to use DLK inhibitor at a lower dose so effects on Jun protein levels are not observed? Perhaps this might support more definitive conclusions about whether inhibiting DLK affects nocodazole-induced activation of p-Jun.

Please note that Figure 1 has quantitative results for Jun in panel 1B not 1D.

3) Quantification for growth cone phenotypes in Figure 1E would be valuable. The authors make the following statement pertaining to this: “By 16 hours after pretreatment with DLK inhibitor, swellings and protrusions appeared along the most distal portions of the axon. In subsequent experiments, we characterized and quantified changes to axonal morphology.” However, the reviewer could not find this quantitative data in Figure 1 or the Supplement. Did reviewer miss this data?

4) Figure 2A notes four treatments but the data in Figure 2B is a comparison of two samples. Presumably one is the double treated sample, but what is the control? This was very hard to follow without figure legends. What are the Dlk associated genes being affected? Could these be annotated with a certain color? Would it be valuable to have two sets of comparisons here to see what genes are upregulated by nocodazole alone, and which effects are reduced when Dlk inhibitor is applied to nocodazole treated cells? This might allow the reader to more easily track which DLK associated genes are affected by nocodazole and how the Dlk inhibitor adjusts their transcriptional profile.

5) While the analysis in Figure 2D-F is quite standard, it did not seem to support a major claim made. Can analysis in F be refined for MAPK activity which was not clearly present? It appeared only “kinase signaling” was a group identified as upregulated.

6) In Figure 4 quantification is shown for filopodial length, but it seems filopodial number is the major phenotype. Would it be valuable to quantify this as well?

7) It appears in Fig 4B and D that Dlk inhibitor application completely removes the presence of all growth cones. Is this the case? Reviewer is concerned that drawing conclusions about how Dlk inhibition affects growth cone size with nocodazole treatment is not valid if there are no growth cones with Dlk inhibitor is applied alone. Would a lower concentration of Dlk inhibitor be helpful?

8) All legends for Figures appear to be absent. Did the reviewer miss these? This created a problem understanding many of the figures and data. At time this also left the reviewer unclear if data shown supports conclusions drawn in text.

Minor comments:

1) Prior studies showed that RhoG/Rac isoform MIG-2 is an upstream genetic activator of DLK. Is it important examine Rac/Rho transcription and whether this changes with nocodazole and Dlk inhibitor treatment? Is this presented in Figure 3A where Rho, Rac and CDC42 signaling is diagramed?

2) Prior studies from invertebrates have demonstrated genetically and biochemically that serine-threonine phosphatases affect Dlk and its downstream signaling. Why did the authors only annotate dual specificity phosphatases? Would it be valuable to annotate data for single specificity serine/threonine phosphatases or tyrosine phosphatases and discussion? Was there a reason the authors only focused on dual specificity phosphatases?

3) The references are not cited consistently between text and references listed. This made it very hard to follow the literature and support for claims made.

4) The Supplementary data is only cited in the Methods. Is this correct?

5) Authors note three papers pertaining to spectraplakin, patronin and RAE1 regulating cytoskeletal instability in the axon. The reviewer did not see data in the papers cited showing Rae1 and axonal microtubule stability. Did the authors miss a citation? Did the reviewer miss a piece of data in published papers?

Reviewer #2: The manuscript by DeVault et al. examines the possibility that the DLK signaling pathway acts as a cytoskeletal stress response (CSR), similar to other homeostatic regulators in other contexts. The authors first provide additional evidence that DLK signaling is activated after cytoskeletal disruption, a finding supported by prior work. They then examine the DLK-dependent transcriptional response signature in neurons exposed to mild cytoskeletal disruption (low-dose nocodazole) and identify upregulated genes that are capable of attenuating DLK signaling and others that are cytoskeletal regulators. They note that this evidence does not conclusively prove their model but is generally supportive.

This study is an important and interesting addition to our understanding of DLK signaling and its findings should be an important starting point for further work. Some clarifications would be useful on certain points, but overall my comments are minor:

1. Some statements e.g. p2 “Within days, injured neurons in vitro return to a basal state” refer to findings from comparatively young, peripheral neurons. It is less clear if this statement holds for central neurons e.g. effects reported in PMID: 28931811 and 31607869 may not be reversible. The original statement should be considered and should likely be reworded.

2. In Fig 1A it appears that total MKK4 is reduced in the presence of DLKi and nocodazole (4th lane). Is this result reproducible and, if so, do the authors have an explanation?

3. It is unclear if the fixation step in the immunostaining protocol (2%PFA) distinguishes microtubules from free tubulin. This is important for findings in Fig 1F, which plot ‘microtubule integrity’ and in the accompanying text. Can the authors provide further evidence to support this conclusion (e.g. from prior studies using this protocol)? If not, the description of these experiments should be rephrased.

4. Do the authors have evidence to rule out that ‘homeostasis’ does not reflect a decrease in nocodazole stability/efficacy at later times (e.g. is the half-life of nocodazole in aqueous solution known, or can ‘aged’ medium from nocodazole-treated cultures induce the same effects in naïve cultures)? This point should be addressed in the Discussion and a short caveat added if needed.

5. The authors initially highlight similarities between injury- and trophic deprivation-induced transcriptional programs (p6) but the outcomes of these two treatments are very different (trophic deprivation causes apoptotic death of young DRG neurons, while injury does not result in in DRG neuron death and later triggers regeneration). This is especially an issue when DUSP upregulation is highlighted as a potential common homeostatic mechanism (p8), because DUSPs are induced by both treatments, despite their different outcomes. The potential roles of DUSP upregulation in the CSR should be considered and, if necessary, this section should be reworded. The same is true for the statement on p10 that ‘DLK activation of DRGs leads to regeneration’ – again, after trophic deprivation-induced DLK activation, death rather than regeneration is the outcome.

6. PLOS authors have the option to publish the peer review history of their article (what does this mean?). If published, this will include your full peer review and any attached files.

Reviewer #1: No

Reviewer #2: No

---

## [Author Response · Author response to Decision Letter 0]

26 Jan 2024

Dear Reviewers, 

We thank you for your comments. All comments are addressed in the document Response to reviewers. 

Best, 

Laura DeVault

---

## [Decision Letter · Decision Letter 1]

29 Feb 2024

The response of Dual-Leucine Zipper Kinase (DLK) to nocodazole: evidence for a homeostatic cytoskeletal repair mechanism

PONE-D-23-37022R1

Dear Dr. DiAntonio,

We’re pleased to inform you that your manuscript has been judged scientifically suitable for publication and will be formally accepted for publication once it meets all outstanding technical requirements. Of the two reviewers that evaluated your original manuscript, one recommended acceptance of the revised version, while the other did not respond in time. The academic editor found that you adequately addressed the comments of both reviewers.

Kind regards,

Gerhard Wiche, Ph.D.

Academic Editor

PLOS ONE

Additional Editor Comments (optional):

Reviewers' comments:

Reviewer's Responses to Questions

**Comments to the Author**

1. If the authors have adequately addressed your comments raised in a previous round of review and you feel that this manuscript is now acceptable for publication, you may indicate that here to bypass the “Comments to the Author” section, enter your conflict of interest statement in the “Confidential to Editor” section, and submit your "Accept" recommendation.

Reviewer #2: All comments have been addressed

2. Is the manuscript technically sound, and do the data support the conclusions?

Reviewer #2: Yes

3. Has the statistical analysis been performed appropriately and rigorously? 

Reviewer #2: Yes

4. Have the authors made all data underlying the findings in their manuscript fully available?

Reviewer #2: Yes

5. Is the manuscript presented in an intelligible fashion and written in standard English?

Reviewer #2: Yes

6. Review Comments to the Author

Reviewer #2: This reviewer thanks the authors for their responses to initial review, in particular their efforts to address the issue of microtubule vs free tubulin immunostaining. The revised manuscript is significantly strengthened and is now suitable for publication.

7. PLOS authors have the option to publish the peer review history of their article (what does this mean?). If published, this will include your full peer review and any attached files.

Reviewer #2: **Yes: **Gareth Thomas

---

## [Editor Report · Acceptance letter]

25 Mar 2024

PONE-D-23-37022R1 

PLOS ONE

Dear Dr. DiAntonio, 

I'm pleased to inform you that your manuscript has been deemed suitable for publication in PLOS ONE. Congratulations! Your manuscript is now being handed over to our production team.

Kind regards, 

on behalf of

Prof. Gerhard Wiche 

Academic Editor

PLOS ONE